# Untapped Potentials of Endophytic Fungi: A Review of Novel Bioactive Compounds with Biological Applications

**DOI:** 10.3390/microorganisms8121934

**Published:** 2020-12-06

**Authors:** Madira Coutlyne Manganyi, Collins Njie Ateba

**Affiliations:** 1Department of Microbiology, North West University Mafikeng Campus, Private Bag X2046, Mmabatho 2735, South Africa; 2Food Security and Safety Niche Area, Faculty of Agriculture, Science and Technology, North West University, Mmabatho, Mafikeng 2735, South Africa; collins.ateba@nwu.ac.za

**Keywords:** bioactive compounds, biological activities, endophytic fungi, discovery, novel

## Abstract

Over the last century, endophytic fungi have gained tremendous attention due to their ability to produce novel bioactive compounds exhibiting varied biological properties and are, therefore, utilized for medicinal, pharmaceutical, and agricultural applications. Endophytic fungi reside within the plant tissues without showing any disease symptoms, thus supporting the physiological and ecological attributes of the host plant. Ground breaking lead compounds, such as paclitaxel and penicillin, produced by endophytic fungi have paved the way for exploring novel bioactive compounds for commercial usage. Despite this, limited research has been conducted in this valuable and unique niche area. These bioactive compounds belong to various structural groups, including alkaloids, peptides, steroids, terpenoids, phenols, quinones, phenols, and flavonoids. The current review focuses on the significance of endophytic fungi in producing novel bioactive compounds possessing a variety of biological properties that include antibacterial, antiviral, antifungal, antiprotozoal, antiparasitic, antioxidant, immunosuppressant, and anticancer functions. Taking into consideration the portal of this publication, special emphasis is placed on the antimicrobial and antiviral activities of metabolites produced by endophytes against human pathogens. It also highlights the importance of utilization of these compounds as potential treatment agents for serious life-threatening infectious diseases. This is supported by the fact that several findings have indicated that these bioactive compounds may significantly contribute towards the fight against resistant human and plant pathogens, thus motivating the need enhance the search for new, more efficacious and cost-effective antimicrobial drugs.

## 1. Introduction

In recent years, there has been a dramatic shift towards a more sustainable, eco-friendly, and natural way of living. Many researchers are of the opinion that there is an alarming increase in drug resistance worldwide and the problem is escalating tremendously, thus rendering the current antimicrobial agents ineffective [1,2,3]. Approximately two million people worldwide are infected with antibiotic-resistant pathogens, resulting in at least 23,000 mortalities per annum [4]. According to the World Health Organization (WHO), antimicrobial resistance (AMR) has emerged as one of the most serious public health concerns of the 21st century [5]. Moreover, due to the many disadvantages and side effects associated with current antimicrobial agents, it is not surprising that a large proportion of individuals, especially those who live in developing countries, are utilizing naturally available bioactive alternatives for their primary healthcare. 

In developing countries, especially those in Africa, Asia, and Latin America, more than 80% of the population utilize medicinal plants to address their primary healthcare needs and wellness [6]. With over 400,000 diverse plant species inhabiting the planet, a majority are capable of treating a broad spectrum of ailments. Due to the fact that they possess a wide variety of biological properties, this has increased interest in research designed to search for “ideal” bioactive compounds that may be of benefit to mankind [7,8,9]. However, over-propagation and excessive usage might lead to endangerment and extinction of the plants. In-depth studies have shown that endophytic fungi are capable of colonizing plant tissues, providing protection, and are a rich source of natural bioactive compounds [10]. 

In addition, the ratio of fungal species to vascular plants is approximately 1:6, which is estimated at 1.5 million fungal species [11]. Endophytes are microorganisms that colonize internal plant tissues without causing any apparent harm to the host plant throughout their life cycle. This interaction is considered as mutualistic symbiosis, whereby both partners benefit from the association [12,13,14]. The endophytes aid in the physiological and ecological roles of the plants, resulting in protection and survival mechanisms. Moreover, endophytic fungi promote growth, prevent abiotic and biotic stresses, such as drought conditions, salinity, extreme temperatures, heavy metal toxicity, and oxidative stress, and provide protection from insect and herbivores [13,15,16]. Endophytes have the ability to prevent resistance mechanisms by overcoming pathogenic invasion using secondary metabolites.

Bioactive compounds are primarily responsible for the beneficial characteristics displayed by endophytic fungi. History has shown that such metabolic compounds might aid in ground-breaking discoveries, e.g., the discovery of penicillin produced by *Penicillium chrysogenum,* which was a milestone in the development of antibiotic drugs, as a “front-line” antibiotic which saved millions of lives; hence, it was referred to as “wonder drug” [17]. Another “gold” bioactive compound is paclitaxel (taxol), which is produced by *Taxomyces andreanae* for chemotherapy purposes [18]. These bioactive metabolites might be structurally classified into alkaloids, benzopyranones, chinones, peptides, phenols, quinones, flavonoids, steroids, terpenoids, tetralones, xanthones, and others [19]. They have exhibited numerous biological properties, including antibacterial, antifungal, immunosuppressants, antiviral, antiparasitic, antioxidant, anti-inflammatory, and anticancer properties [20]. 

New biotechnological advances concentrate on the search for and utilization of novel bioactive compounds extracted from endophytic fungi. Despite all of this, only a tiny portion of endophytic fungi have been isolated and investigated for their biological activities. In this current review, we focus in depth on the various biological properties demonstrated by endophytic fungi. We also identify novel bioactive compounds associated with the activities. Furthermore, we shed light on the activities against resistant pathogenic microorganisms. Unexplored niche areas such as these will progress the drug development process tremendously. Endophytic fungi are an abundant source of natural bioactive compounds which are novel, renewable, and low in toxicity and more efficacious, more potent, more affordable, safer, and less resistant than conventional antimicrobial agents. Hence, this will alleviate the massive burden on public healthcare systems and aid in the medical and pharmaceutical industries. 

## 2. Biological Properties of Novel Bioactive Compounds from Endophytic Fungi

Various biological activities, such as antibacterial, antifungal, immunosuppressant, antiviral, antiparasitic, antioxidant, anti-inflammatory, and anticancer, were exhibited by endophytic fungi with a wealth of bioactive metabolites. In this section, we investigate endophytic fungi as a source of countless bioactive compounds which might be beneficial for human health. 

### 2.1. Bioactive Compounds as an Alternative Antibacterial Agents

Since the golden era of penicillin, antibiotics have become a staple conventional medicine and have reduced the fatality rate by saving millions of lives, as was proven in World War II [21]. In the 1940s, it was the first prescribed antibiotic utilized in the treatment of life-threatening infections. Shortly afterward, the development of antibiotic resistance, and particularly the detection of multiple antibiotic-resistant (MAR) bacteria, has resulted to downturn reduction of the activity of antibiotics [22]. The dramatic increase in microbial resistance reduces the efficacy of existing antibiotics and, therefore, negatively affects their applications in human medicine. Over the last century, a number of antibiotics have emerged in the market, and despite this developmental processes, there is evidence of antibiotic resistance being developed against all these antibiotics (Figure 1) [23]. According to the World health Organization (WHO), antibiotic resistance remains one of the leading threats to human health and poses a severe financial burden on healthcare systems worldwide [24]. 

Aharwal and co-workers [25], recognized the severity of antibiotic resistance and emphasized the importance of innovative research and development strategies. Several studies have established that endophytes have resistance mechanisms in place to overcome pathogenic invasion through the production of secondary compounds [15,19,26]. Bioactive metabolites are low-molecular-weight, organic natural substances produced by microorganisms that possess activities at low concentrations against other microorganisms, in respect to bioactive compounds with potential antibiotic properties. Antibacterial compounds produced by endophytes have been shown to occupy a broad spectrum of structural classes, such as alkaloids, peptides, steroids, terpenoids, phenols, quinines, and flavonoids [27]. 

Recently, numerous scientists have demonstrated that endophytic fungi have a variety of beneficial, novel, and effective bioactive metabolites possessing antibacterial activity. Endophytic *Alternaria alternate* (AE1) was isolated from healthy and mature leaves of *Azadirachta indica* plants located in Santiniketan. The fungal extracts exhibited excellent antimicrobial activities against both Gram-positive and Gram-negative bacteria (*Bacillus subtilis* MTCC 121, *Listeria monocytogenes* MTCC 657, *Staphylococcus aureus* MTCC 96 and *Staphylococcus epidermidis* MTCC 2639, *Salmonella typhimurium* MTCC 98, *Pseudomonas aeruginosa* MTCC 741, and *Escherichia coli* MTCC 1667) with minimum inhibitory concentration (MIC) values of 300–400 μg/mL. The mode of action was determined as being cidal and further analysis demonstrated that cell lysis or leakage of cell membrane was occurring. Gas chromatography–mass spectrometry (GC-MS) analysis confirmed that the fungal extracts consist of several bioactive compounds [28]. The bioactive compounds produced from endophytic fungal extracts of *Penicillium* sp. demonstrated strong effectiveness against *Enterococcus faecalis* with an MIC value of 62.50 mg/mL [29]. Linoleic acid (9,12-octadecadienoic acid (Z,Z)) and cyclodecasiloxane produced by endophytic *Alternaria* sp. were successfully isolated from native South African *Pelargonium sidoides*, and this is considered as a first report. These fungal metabolites inhibited several food-borne and food spoilage bacteria, including *Bacillus cereus, Escherichia coli,* and *Enterococcus faecium,* and *E. gallinarum* showed a 2–12-mm zone of inhibition [30]. Despite the fact that root-knot nematodes such as *Meloidogyne incognita* are persistent parasites in plants and contribute to about 5% of global crop losses [31], a biocontrol investigation revealed that the endophytic fungus *Piriformospora indica* was capable of inhibiting the root-knot nematode parasite and, thus, enhanced plant growth [31].

Kjer et al. (2009) discovered two new secondary metabolites (10-oxo-10H-phenaleno [1,2,3-de] chromene-2-carboxylic acids and xanalteric acids I and II, Figure 2) which were produced by the fungus *Alternaria* sp., isolated from the mangrove (*Sonneratia albacollected*) plant located in China. The metabolites exhibited strong antibacterial properties against *Enterococcus faecalis, Pseudomonas aeroginosa*, and *Staphylococcus epidermidis* [32]. 

Ding and colleagues (2019) also reported compounds 1 and 2 as novel isocoumarin derivatives (Figure 2) with a distinctive butanetriol group at C-3 using NMR and MS. Furthermore, the bioactive compounds exhibited some activity against Gram-negative bacteria. Compounds 1 and 2 showed effectiveness against *E. coli*, with MIC values of 32 μg/mL [34]. Besides that, other studies also discovered novel ester metabolites isolated from endophytic fungus from the eastern larch that possess antibacterial efficacy against *Vibrio salmonicida, Pseudomonas aeruginosa,* and *Staphylococcus aureus*. These compounds were identified as 8,1′,5′-trihydroxy-3′,4′dihydro-1′H-[2,4′]binaphthalenyl-1,4,2′-trione, (1), and 2-methyloctanoic acid 6-oxo-2-propenyl-3,6-dihydro-2H-pyran-3-yl ester, (2) [35]. Figure 2 represent some bioactive compounds are with antibacterial effect.

### 2.2. Bioactive Compounds as an Alternative to Antifungal Agents

The diversity of fungal species is currently estimated at 2.2 to 3.8 million worldwide, with soil fungal populations contributing a significant portion, and fungi are regarded as ubiquitous [36]. This suggests that there might be more fungal interaction, whether beneficial or pathogenic. Recently, fungal diseases were accountable for over a 1.6 million mortality rate per annum, with over one billion individuals in a state of severe morbidity. In the agricultural sector, fungal pathogens might lead to damage or loss of crops, economic losses, and, eventually, affect food security and food production [37]. In addition, there has been a dramatic increase in the resistance of microorganisms to current antimicrobial agents, including antifungal agents. Despite the inefficacy of antifungal agents, there is currently a lack of therapeutic agents on the market. Fungal diseases are generally underrated and overlooked, although they cause secondary infections to hospitalized patients. Pathogenic fungi are opportunistic and affect immunocompromised individuals, thus posing a serious burden on current health care systems worldwide [38]. 

Research advances in modern antifungal agents rely highly on screening for novel bioactive compounds that have gained popularity in the drug development process and will aid in the fight against animal and human fungal pathogens. Peláez and co-workers (2000) established that new triterpene glycoside exhibited good antifungal inhibitory activities against *Candida* and *Aspergillus* sp. with 19 and 30 mm, respectively. Furthermore, in vivo studies conducted on mouse models showed moderate efficacy against candidiasis [39]. Recent studies revisited formerly undescribed bioactive compounds—strobilurin G, favolon, pterulinic acid, and 2,3-dihydro-1-benzoxepin derivative isolated from *Favolaschia calocera*—which displayed antifungal effectiveness. The minimum inhibitory concentration (MIC) displayed by compound 1 was ≤ 9.37 and ≤ 18.75 μg/mL against *Candida tenuis* and *Mucor plumbeus*, respectively [40]. Similar studies showed that endophytic fungi (*Penicillium* sp. (C7B) and *Trichoderma* sp. (B1C, C4E, C4D)) were effective against *Escherichia coli, Staphylococcus aereus*, and *Vibrio alginolyticus*, displaying clear-zone diameters of 17.91 ± 0.84 mm; 17.78 ± 0.83 mm; 17.66 ± 0.83 mm; 16.72 ± 1.15 mm, and 13.65 ± 0.27 mm, respectively [41].

“Pestalachlorides A” (C21H21Cl2NO5) and “B” (C20H18Cl2O5) are chlorinated benzophenone derivatives that inhibited plant pathogenic fungi, *Fusarium culmorum, Gibberella zeae,* and *Verticillium albo-atrum*, exhibiting MIC activities of 7.2, 144.4, and 114.4 Μm respectively [42]. Well known compounds also include iridoide, glcopyranoside and saponin (Figure 3).

Endophytic *Humicola* sp. (JS-0112 strain) proved to be an ideal candidate for the development of novel fungicides by controlling plant pathogenic *Sclerotinia homoeocarpa* [43]. Current studies have discovered two novel polyketides with an unprecedented “C_12_-C_6_” carbon skeleton isolated from endophytic *Phomopsis* sp. (CFS42). Moreover, the compounds inhibited activities against *Bipolaris sorokinian*, *Alternaria alternat*, *Fursarium avenaceum*, and *Curvularia lunata*. Therefore, this reiterates the untapped wealth of novel bioactive compounds produced by endophytic fungi [44].

### 2.3. Bioactive Compounds for Treating Cancer Cells (Anticancer Activity)

According to the World Health Organization (WHO), cancer is a group of diseases caused by the malignant growth of cells or tumor resulting from uncontrolled cell division [45]. Cancer was primarily responsible for one in six deaths in among humans in annum of 2018, resulting in an overall estimate of 9.6 million deaths worldwide. Cancer was reported to be the second leading cause of deaths worldwide. Men are often more susceptible to lung, prostate, colorectal, stomach, and liver cancer, while women often suffer from breast, colorectal, lung, cervical, and thyroid cancer [45]. The majority (70%) of deaths caused by cancer occur in low- and middle-income countries in Africa, Asia, and Central and South America [45]. Currently, therapeutic processes for treating cancer patients present tremendous challenges to both the physician and the patient, resulting from the lack of precision as well as lack of bioavailability, since most anticancer agents are naturally lipophilic and have a high first-pass effect. In addition, these agents are also non-specific towards their targets and, therefore, may interact with non-cancerous tissues. Moreover, cancer patients most often experience adverse effects after therapeutic procedures that are also associated with high toxicity [46,47,48]. Due to these complications, cancers and their associated diseases are considered to be a serious healthcare burden to patients globally [49]. This explains the reasons for continuous focused investigations aimed at discovering new natural bioactive compounds, especially from endophytes, that may serve as alternative agents to combat cancer [47,48].

The ground-breaking study resulting in the discovery of paclitaxel, also known as the “golden” compound, produced renewed hope in the search for novel anticancer agents, and the compound gained significant popularity because of its distinctive mode of action when compared to other anticancer agents. Paclitaxel obstructs the uncontrolled duplication of cancerous cells, thereby reducing their growth and spread. Paclitaxel with a chemical formula of C_47_H_51_NO_14_ is formulated from ‘taxol”, the first anticancer drug to generate billions of dollars. The endophytic fungus (*Taxomyces andreanae*) isolated from the Pacific yew bark (*Taxus brevifolia*) produces the anticancer bioactive compound paclitaxel. A number of studies have also isolated taxol and its related compounds from a variety of endophytes, including *Grammothele lineata* [50], *Aspegillus aculeatinus* [51], *Alternaria brassicicola* [52], and *Acremonium, Colletotrichum*, and *Fusarium* spp., from an ecologically altered *Taxus baccata* [53]. Taxol and its associated compounds were characterized and confirmed using UV absorption, HPLC, FTIR spectra, and LC–ESI–MS [50,51,52,53]. Despite its very effective anticancer properties, taxol is rare, which amplifies the need to enhance the search for alternative sources to this very important bioactive metabolite [51]. In addition, plants that produce taxol appear to be “rare”, which, coupled with the fact that the production of taxol from *Taxus* plants is a complex process, means that an alternative will be to constantly assess taxol-producing endophytic fungi due to the ease and practicality of obtaining these valuable compounds. In order to understand the complexity of the processes or molecular mechanisms involved in the production of taxol, [51] assessed the transcriptome of *Aspergillus aculeatinus* Tax-6, indicating that genes involved in two pathways (mevalonate (MVA) and nonmevalonate (MEP)) need to be expressed. Despite the fact that the potential of endophytes to produce taxol varies, with *Grammothele lineata* producing 382.2 μg/L [50], *Alternaria brassicicola* producing 140.8 μg/L taxol [52], and *Acremonium, Colletotrichum*, and *Fusarium* spp., which also produced up to 116.19 μg/L [53], the production of a very high yield (560 µg/L) by the mutant strain *A. aculeatinus* BT-2 when compared to the wild-type *A. aculeatinus* Tax-6 strain [51] indicates the importance of genetic manipulation in the search for these very important bioactive compounds. In addition to the previously mentioned studies, other endophytic fungi, such as *Pestalotiopsis microspore, Alternaria alternata, Periconia* sp., *Pithomyces* sp., *Chaetomella raphigera, Monochaetia* sp., *Seimatoantlerium nepalense*, *Botryodiplodia theobromae*, *Phyllosticta spinarum, Pestalotiopsis terminaliae*, and *Bartalinia robillardoides*, have also been found to produce anticancer bioactive agents that chemically belong to taxols [48,54]. Several taxol-producing endophytes therefore provided a cheaper and more accessible alternative for anticancer agents [54], and it is on this basis that taxol compounds were approved by the Food and Drug Administration (FDA) for the treatment of advanced breast cancer, lung cancer, and refractory ovarian cancer [55].

In addition to paclitaxel, camptothecin (C_20_H_16_N_2_O_4_), a bioactive alkaloid compound initially isolated from *Camptotheca acuminata* wood in China, also exhibited anticancer properties [56,57]. Furthermore, another anticancer compound, “chaetoglobosin U”, was produced by the fungus *Chaetomium globosum*, which lives synergistically in the stem of healthy *Imperata cylindrica* [58]. Other novel bioactive compounds, such as cytoglobosins C17 and D18 alkaloids from the endophytic fungus *Chaetomium globosum*, inhibited antitumor activities on the cancer cell line A549 [59]. Several bioactive compounds have been shown to possess anticancer activities (Figure 4). Li et al. (2013) [58] also established that the endophyte *Chaetomium globosum* from the *Ginkgo biloba* plant contained three novel compounds that include azaphilone alkaloids, chaetomugilides A–C, and chaetoviridin E, all of which displayed high cytotoxic activities against the human cancer cell line HePG2 [58]. Other endophytic fungi belonging to the genera *Xylaria, Phoma, Hypoxylon,* and *Chalara* have also been reported to produce cytochalasins, which possess antitumor activities. In addition, three novel cytochalasins, namely cytochalasin H, cytochalasin J, and cytochalasin E, were also extracted from *Rhinocladiella* sp. isolated from *Tripterygium wilfordii* [60]. Podophyllotoxin (C_22_H_22_O_8_), a lignan-type bioactive compound which was originally isolated from *Podophyllum peltatum* L. in the 1880s, and its derivatives are well-known to possess a variety of biological properties and are used as cathartic, purgative, antiviral, vesicant, antibacterial, antihelminthic, and antitumor agents. Etoposide and teniposide are chemotherapeutic medications produced from podophyllotoxin, and they are currently available in the market. In addition, endophytes *Fusarium oxysporum, Aspergillus fumigatus, Phialocephala fortinii,* and *Trametes hirsute*, as well as some belonging to the genera *Trichoderma, Penicillium*, and *Phomopsis*, have also been reported to produce the compound podophyllotoxin, which exhibits anticancer activities [61,62]. 

The discovery of other lead compounds was intensified by the isolation of camptothecin from the endophytic fungi *Fusarium solani* housed in *Camptotheca acuminata* Decaisne (Nyssaceae) wood located in China. Camptothecin (C_20_H_16_N_2_O_4_) is an alkaloid topoisomerase compound that displayed remarkable potent antineoplastic efficacy. Camptothecin and 10-hydroxycamptothecin were utilized in the development of the drugs topotecan and irinotecan, which are chemotherapeutic. Since then, precursor 9-methoxycamptothecin and 10-hydroxycamptothecin have been reported to exhibit powerful anticancer properties [63,64]. In addition to these findings, some reports have revealed that camptothecin and methoxy camptothecin are capable of enhancing the development of fruits and seed germination phases, thus assisting in protection against seed-borne pathogens [65].

Another significant group of bioactive compounds are phenylpropanoids, which are naturally synthesized by plants. However, research has shown that endophytes also produce phenylpropanoids. These compounds belong to the largest group of secondary metabolites that possess an aromatic ring with a 3-carbon propene tail, thus resulting in a C6-C3 carbon skeleton. These compounds are from the amino acids phenylalanine and tyrosine [66]. In cancerous cells, phenylpropanoids prevent overexpression of histone deacetylase (HDAC), thus inhibiting the cell cycle and inducing apoptosis [66]. Phenylpropanoid derivatives possessing antimicrobial activities have been extracted from the endophytic fungus *Aspergillus* sp. (ZJ-68), associated with mangrove [67]. In another study, fusarubin and anhydrofusarubin isolated from *Cladosporium* species inhibited cell growth and induced apoptosis of human cancer cell lines HL-60, U937, and Jurkat [68]. Although both compounds significantly increased apoptosis of these cancerous cells with increase in concentrations, fusarubin significantly decreased the percentage of cells in the S phase while increasing those in the G2/M phase [68]. On the contrary, anhydrofusarubin increased the percentage of cells in the G0/G1 phase but decreased those in the S and G2/M phases [68]. These findings provide a valid basis for the need to focus on investigations aimed at constantly assessing the potential of endophytic fungi to produce bioactive compounds, given the evidence that they have an array of metabolites, such as alkaloids, macrolides, terpenoids, flavonoids, glycosides, xanthones, isocoumarins, quinones, phenylpropanoids, aliphatic metabolites, and lactones, with powerful anticancer properties [69]. Nonetheless, only a small proportion of endophytes have been investigated so far, thus requiring more research in the niche area. 

### 2.4. Bioactive Compounds as a Potential Antioxidant Agent

Oxidation is a chemical reaction resulting in the loss of electrons from an atom, and this may produce free radicals. Naturally occurring free radicals are unstable molecules produced during chemical reactions such as digestion. These free radicals may participate in chain reactions which might potentially cause cell damage in the human body [70,71]. This is mainly due to the fact that after an atom losses an electron, the cell becomes imbalanced, thus resulting in cell damage. Hence, when exposed to oxidative stress, cells may suffer from a wide range of diseases that may also include chronic complications in humans [72]. Extensive studies have demonstrated that exposure of cells to oxidative stress contributes to cellular degeneration, cancer, atherosclerosis, coronary heart ailments, diabetes, Alzheimer’s disease, and hepatic and kidney damage as well as other neurodegenerative disorders [73]. Some severe side effects of oxidative stress and their associated diseases in humans are listed in Table 1. 

Antioxidant agents are utilized to combat, prevent, and treat diseases that are linked with the presence of reactive oxygen species (ROS), and these agents have displayed very high efficacy against damage caused by ROS. ROS have been reported to boost the immune system by enabling cell signaling to occur. Various industries, such as the food, pharmaceutical, and agricultural sectors, use antioxidant compounds for beneficial purposes. Irrespective of the health concerns associated with oxidative stress, the search for safer, more efficacious, and cost-effective natural antioxidants is highly anticipated. Novel natural bioactive compounds have been reported to serve as a shelter against oxidative damage by preventing or reducing free radicals and reactive oxygen species. Various studies have demonstrated that molecules, including phenolic acids, phenylpropanoids, and flavonoids, lignin, melanin, and tannins, exhibited antioxidant activity [74,75]. There is considerable evidence that endophytic fungi produce several antioxidant compounds that are responsible for the stress tolerance in host plants. The endophytic fungus *Fusarium oxysporum* from the leaves of *Otoba gracilipes* exhibited antioxidant activity with as much as 51.5% of a scavenging effect on 2,2-diphenyl-1-picrylhydrazyl (DPPH) after 5 min of reaction [76]. A total of forty-one bioactive compounds from the endophyte *Xylaria* sp. were isolated from the medicinal plant *Ginkgo biloba*, and these compounds displayed antibacterial, antioxidant, anti-cardiovascular, anticancer, and antimicrobial properties. Phenolic and flavonoid compounds, among others, have been shown to possess very effective antioxidant properties [77]. Recently, studies conducted on the endophytic fungus *Alternaria alternata* AE1 isolated from *Azadirachta indica* also revealed that it produced secondary metabolites that possess very effective antioxidant properties [28]. DPPH free radical and superoxide radical scavenging tests of the secondary metabolites displayed antioxidant potentials with an IC_50_ value of 38.0 and 11.38 μg/mL, respectively [28]. Chemical characterization of methanol extracts of two filamentous fungal strains revealed the presence of the residues chlorogenic acid, neochlorogenic acid, rutin, and quercetin 3-acetyl-glucoside (Figure 5), and the fungal extracts displayed significant antioxidant activities [78]. In addition, another investigation identified the biomolecules pestacin, isopestacin, and 1,3-dihydro isobenzofurans from the endophytic fungus *Pestalotiopsis microspore* housed in *Terminalia morobensis*, which also displayed very effective antioxidant activities [79].

Polysaccharides produced by plants and microorganisms have been widely studied due to their natural antioxidants properties. Three polysaccharides, namely exopolysaccharide (EPS), water-extracted mycelial polysaccharide (WPS), and sodium hydroxide-extracted mycelial polysaccharide (SPS), produced by *Fusarium oxysporum* Dzf17 from *Guazuma tomentosa* displayed antioxidant properties [80]. In addition, total phenol and flavonoids detected from the culture filtrate of *Phyllosticta* sp. displayed significant antioxidant properties. When subjected to the 2,2′-azino-bis(3-ethylbenzothiazoline-6-sulfonic acid (ABTS) and DPPH radical assays, these compounds exhibited EC_50_ values of 580.02 and 2030.25 μg/mL, respectively [81]. 

Novel trichothecene macrolides (Figure 5) produced by the endophytic fungus *Nerium oleander* L (Apocynaceae) isolated from *Trachelospermum jasminoides* also displayed antioxidant potentials [82]. Using chemical analysis, six new macrolides comprising myrothecines D–G (1–4), 16-hydroxymytoxin B, and 14′-dehydrovertisporin and four 10,13-cyclotrichothecane derivatives were detected in three endophytes of *Myrothecium roridum* IFB-E008, IFB-E009, and IFB-E012. In addition, an investigation into their antimicrobial properties revealed that the compounds exhibited sufficient cytotoxicity based on their bioactive data [83].

### 2.5. Bioactive Compounds for Treating Infectious Parasites

Parasitic infections in humans are caused by protozoa, helminths, and ectoparasites that live on or in a host organism and make use of the host resources for their survival. Disease-causing parasites are known to contribute significantly to the rate of morbidity and mortality in humans worldwide, especially in developing countries that have a large proportion of vulnerable populations [84]. This, therefore, presents significant challenges to already overburdened public healthcare systems, thus resulting in huge economic losses. Approximately 48.4 million cases of parasitic disease resulting in one million deaths are reported annually [84]. Despite this, there are limited highly effective antiparasitic drugs current available in the market, especially given the challenges faced with resistance of parasites to the drugs [85]. Moreover, there is evidence which indicates that parasitic organisms are rapidly developing resistance against anti-parasitic drugs and the resistant strains are spreading at an alarming rate. This, therefore, calls for the need to intensify the search for novel, more potent, and less toxic compounds that may be more effective against these pathogens [86,87,88]. However, there is also substantial evidence which indicates that endophytes possess a pool of novel bioactive compounds that might be very useful in the discovery of anti-parasitic drugs. 

*Diaporthe phaseolorum*-92C (92C), an endophytic fungus that inhabits the roots of *Combretum lanceolatum*, displayed significant anti-parasitic activity against *Trypanosoma cruzi* by reducing up to 82% of the number of amastigotes and trypomastigotes. The bioactive molecule 18-des-hydroxy Cytochalasin H exhibited nematocidal activity and reduced the viability of promastigotes of *Leishmania amazonenses* with an IC_50_ of 9.2 μg/mL [89]. Another study demonstrated that oxylipin (9Z,11E)-13-oxooctadeca-9,11-dienoic acid obtained from fungal extracts of the endophytic fungus *Penicillium herquei* strain BRS2A-AR was potent against *Plasmodium falciparum* 3D7, *Trypanosoma brucei, Leishmani donovani*, and *Leishmania* sp., with IC_50_ values higher than 100 μM, therefore displaying very excellent anti-parasitic activities [90]. *Alternaria alternata* P1210 from the roots of the halophyte *Salicornia* sp. produced two new biosynthesized dimeric compounds belonging to the class alternariol, namely (±)-alternarlactones A and B. Preliminary results revealed that these compounds possessed anti-parasitic potentials [91].

Mao and colleagues (2019) discovered two novel decalin/tetramic acid hybrid metabolites—hyalodendrins A and B, isolated from the endophytic fungus *Hyalodendriella* sp. Ponipodef12 using spectroscopic chemical analysis. These compounds were capable of inhibiting the growth of fourth-instar larvae of *Aedes aegypti* [92]. Bioassay was carried out using one hundred and fifty-two (*n* = 152) endophytic fungi to determine their anti-plasmodial activity using a 96-well microtiter plate. The results showed that extracts from *Fusarium* sp. AMst1 (IC_50_ = 1.16–1.43 µg/mL), *Trichoderma afroharzianum* AMrb7 (IC_50_ = 1.71–2.31 µg/mL), and *Penicillium tropicum* AMb3 (IC_50_ = 1.90 µg/mL) possess significantly effective anti-plasmodial activities against *Plasmodium falciparum* strains [93]. The findings of another study in Brazil demonstrated that the endophyte *Phyllosticta capitalensis* from *Tibouchina granulosa* (Desr.) Cogn. (Melastomataceae) inhibited *Leishmania* species and *Trypanossoma cruzi*. Based on high-resolution mass spectrum analysis (UHPLC-HRMS), 18 compounds were identified from *P. capitalensis* and crude extracts from the endophyte displayed growth inhibitory activities against *Leishmania amazonensis*, *L. infantum*, and *Trypanosoma cruzi* with IC_50_ values of 17.2, 82.0, and 50.13 μg/mL, respectively. Given that the diseases Leishmaniasis and Chagas are abandoned tropical diseases caused by protozoa and infect over 12 million individuals globally, these findings present a significant hope for mankind [94]. Citrinin, cochiloquinone A, palmarumycin CP_18_ and others (Figure 6) are amongst compounds exhibiting antiparasitic activities.

### 2.6. Bioactive Compounds with the Potential of Serving as Immunosuppressive Drugs

Immunosuppressive drugs, also known as antirejection medications, are utilized to suppress, reduce, or prevent allograft rejection during organ transplant in patients [95]. They therefore play a significant role in the treatment of autoimmune disorders, such as rheumatoid arthritis, lupus, psoriasis, and insulin-dependent diabetes [95]. Currently, the effectiveness of immunosuppressive drugs is affected by a number of side effects, and given that their demand is high, there is a need to hasten the search for safer but more reliable drugs in order to alleviate these problems. A number of studies have established that endophytes are capable of producing bioactive molecules with immunosuppressive potentials [96,97]. Data generated using chemical analysis revealed the presence of a novel amide derivative (-)mycousnine enamine biomolecule that was produced by the endophyte *Mycosphaerella nawae* ZJLQ129, isolated from *Smilax china* leaves [98]. Furthermore, cyclosporin A and (-)mycousnine enamine have selectively inhibited T cell proliferation by blocking the expression of the surface activation antigens CD25 and CD69. These findings confirm that endophytic fungi may serve as a potential source for potent immunosuppressants that have low toxicity but high selectivity [98]. In addition to these, a total of nine polyketides, consisting of two novel benzophenone derivatives, peniphenone and methyl peniphenone, and seven known xanthones (Figure 7), were extracted from the endophytic fungus *Penicillium* sp. ZJ-SY2, which was associated with mangrove *Sonneratia apetala* leaves. These compounds exhibited excellent immunosuppressive properties, with IC_50_ values ranging from 5.9 to 9.3 μg/mL [97]. *Xylaria longipes* HFG1018, obtained from the basidiomycete *Fomitopsis betulinus*, which is associated with rotting of wood, produced eighteen new nor-isopimarane diterpenes, xylarinorditerpenes A–R (1–18), some of which possessed immunosuppressive potential [99]. 

### 2.7. Bioactive Compounds with Antiviral Properties

Viruses are microorganisms that multiply only within living cells and are a leading cause of mortality and morbidity in humans globally. Current antiviral drugs and vaccines are crucial in combating life-threatening diseases in humans [100]. In addition, the emergence of resistance of viruses to the available antiviral drugs reduces the efficacy of their therapeutic potentials, thus resulting in a severe public health concern globally. In effect, ideal antiviral drugs should be potent against the target viral strains but with minimal side effects to the host cells. This therefore affirms that the modes of action of antiviral drugs are usually directed at preventing or inhibiting the infection by targeting viral proteins or the host cellular factors that viruses exploit in order to reproduce and gain control of cellular processes [101,102]. To address this problem, the search for and the discovery and development of new, cost-effective, and more potent antiviral drugs as well as vaccines is mandatory. Studies have been carried out to assess the potential of endophytes to produce promising natural bioactive compounds with antiviral properties [102]. 

Isoindolones compounds, namely emerimidines A and B, emeriphenolicins A and D, as well as other compounds, including aspernidines A and B, austin, austinol, dehydroaustin, and acetoxydehydroaustin, were discovered in the endophytic fungus *Emericella* sp. (HK-ZJ) from the mangrove plant *Aegiceras corniculatum.* A bioassay using the cytopathic effect (CPE) test revealed that the fungal extracts displayed potency against influenza A viral (H1N1) [103]. Fungal extracts of *Nigrospora sphaerica* (No.83-1-1-2), *Alternaria alternata* (No.58-8-4-1), and *Phialophora* sp. (No.96-1-8-1) exhibited some antiviral activity against herpes simplex virus (HSV). Extraction and identification of compounds revealed two novel heptaketides, (+)-(2S,3S,4aS)-altenuene (1a) and (−)-(2S,3S,4aR)-isoaltenuene, along with six recognizable compounds, (−)-(2R,3R,4aR)-altenuene, (+)-(2R,3R,4aS)-isoaltenuene, 5′-methoxy-6-methyl-biphenyl-3,4,3′-triol, alternariol (4), alternariol-9-methyl ether, and 4-hydroxyalternariol-9-methyl ether [104].

Recently, endophytic fungi were isolated from medicinal plants of Egyptian origin, which displayed significant antiviral properties against herpes simplex (HSV-2) and vesicular stomatitis viruses (VSV) [105]. It was identified that the endophyte *Pleospora tarda* was responsible for the potent antiviral agents classified as alternariol and alternariol-(9)-methyl compounds [105]. Lui et al. (2019) identified a new rare 14-nordrimane sesquiterpenoid, extracted from the endophyte *Phoma* sp., isolated from the roots of *Aconitum vilmorinianum* [106]. The compounds also inhibited the growth of influenza A virus (A/Puerto Rico/8/34, H1N1). Other bioactive compounds with antifungal activities included (–)-6-methoxymellein, 7-hydroxy-3, 5-dimethyl-isochromen-1-one, norlichexanthone, 6-methylsalicylic acid, and gentisyl alcohol [106]. In addition, the hydroanthraquinone (Figure 8) derivative, 6-O-demethyl-4-dehydroxyaltersolanol A, azaphilones, 8,11-didehydrochermesinone B, and (7S)-7-hydroxy-3,7-dimethyl-isochromene-6,8-dione have recently been identified as compounds from the culture extract of *Nigrospora* sp. YE3033 which resides in the plant *Aconitum carmichaeli*. A preliminary bioassay indicated that these compounds displayed strong antiviral activity against the influenza viral strain A/Puerto Rico/8/34 (H1N1) [107]. 

An investigation of endophytes associated with *Penicillium* sp. FKI-7127 led to the detection of the brefeldin A compound (Figure 8), exhibiting potent antiviral properties [108]. Several endophytic fungal strains, including *Fusarium equiseti, Scopulariopsis fusca*, and *Geotrichum candidum*, were obtained from brown alga *Padina pavonica*, located in the Red Sea. Out of these fungi, *F. equiseti* exhibited the highest antiviral activity against hepatitis C virus (HCV) NS3-NS4A protease, with an IC_50_ of 27.0 µg/mL. Structural characterization using MS and NMR spectral analysis of the metabolites from this endophyte showed the presence of two diketopiperazines (Figure 8) (cyclo-L-AlaL-Leu and cyclo(L-Tyr-L-Pro)) and two nucleosides (cordycepin and Ara-A) [109]. To date, more than one hundred endophytic fungi isolated from desert plants have been established to possess potent antiretroviral activities against human immunodeficiency virus type 1 (HIV-1). The extracts from these fungi demonstrated less than 30% cytotoxic activities in T lymphocytes [110]. Three unidentified chromanones were extracted from the fungal strain *Phomopsis* sp. CGMCC No. 5416, obtained from the stems of *Achyranthes bidentata*. These compounds displayed promising antiviral activities against HIV-1 [111]; we therefore suggest that fungal secondary metabolites may serve as important sources for discovering new antiviral drugs or lead compounds. 

### 2.8. Bioactive Compounds as Potential Antitubercular Drugs

Tuberculosis (TB) is a life-threatening disease usually affecting the lungs and is caused by the bacteria *Mycobacterium tuberculosis*. The risk of contracting TB is significantly higher among individuals whose immune system has been compromised and children under the age of 5 years, particularly in developing countries, such as China, Indonesia, the Philippines, Pakistan, Nigeria, Bangladesh, and South Africa [112]. Infectious TB is most often associated with high fatality rates in humans worldwide [112,113]. Given that TB is more common in people with compromised immune systems, there is evidence of co-infection between TB and HIV in patients, and the World Health Organization (WHO) reported that of the 1.5 million patients who died from TB in 2018, 251,000 were also infected with HIV [112]. In addition, the multi-drug resistance of *Mycobacterium tuberculosis* against available drugs is increasing at an alarming rate and has eventually become an issue of severe public health concern [113]. Approximately 484,000 new cases of resistance to the first-line antimicrobial drug rifampicin, which was regarded as the most potent drug, were reported, which is now a severe public health threat [114] and, thus, requires urgent attention. Nonetheless, tuberculosis is curable and preventable; thus, the constant search for natural bioactive compounds, especially from endophytes, now paves the way for the discovery of new, more effective, alternative agents to combat tuberculosis. *Gliocladium* sp. MR41 was capable of producing polyols 3 and 4 (Figure 9), compounds that displayed inhibitory activities against *M. tuberculosis* at a minimum inhibitory concentration (MIC) of 0.78 µg/mL [115]. Other compounds with potential antituberculosis activities include phomoenamide, abyssomicin, tenuazonic acid and phomonitroester (Figure 9).

These were the first reports of such compounds being produced by a fungus [115]. *Glycyrrhiza glabra* L. plant cultivated in the Kashmir Himalayas harbored a wide diversity of endophytic fungi that comprised, but were not limited to, *Fusarium oxysporum* strain (KT166447) and *Colletotrichum gleosporoides* strain (KT166445), which displayed strong inhibitory potentials against *Mycobacterium tuberculosis* (M. tb) strain H37Rv, with MIC values of 18.5 and 75 µg/mL, respectively [116]. In a recent study involving a review of findings between 2014 and 2015 detailing various biological activities, including antituberculosis, and that endorses fungal strains as a source of endless bioactive compounds, it was revealed that endophytes provide a renewed hope for the detection and development of potential antitubercular drugs [117]. Table 2 listed several endophytic fungi isolated from various host plants that were capable of producing compounds with several bioactive activities. 

## 3. Conclusions

Currently, we are losing the fight against ineffective, toxic, and expensive therapeutic antimicrobial drugs. However, endophytes provide a suitable alternative since they are a warehouse filled with novel bioactive compounds with endless possibilities of biological properties. Over the past few years, endophytic fungi have attracted tremendous attention in the drug development process as they are ubiquitous and abundantly availability. Numerous studies have reported novel, beneficial bioactive compounds exhibiting biological properties, such as antibacterial, antidiabetic, antifungal, anti-inflammatory, antiprotozoal, antituberculosis, insecticidal, immunomodulatory, antiviral, anticancer activities, anthelmintic, etc., that were successfully isolated from endophytic fungi. Despite this, limited research has been conducted on the valuable bioactive compounds from endophytic fungi. Research priorities need to shift towards biotechnological advances to accelerate the screening of new biomolecules for the treatment of numerous life-threatening diseases, thus safe-guarding human health. All things considered, an untapped wealth of novel bioactive compounds resides within endophytes, thus ensuring the discovery of new bioactive compounds for potential applications in the agricultural, food, medical, and pharmaceutical industries.

## Figures and Tables

**Figure 1 microorganisms-08-01934-f001:**
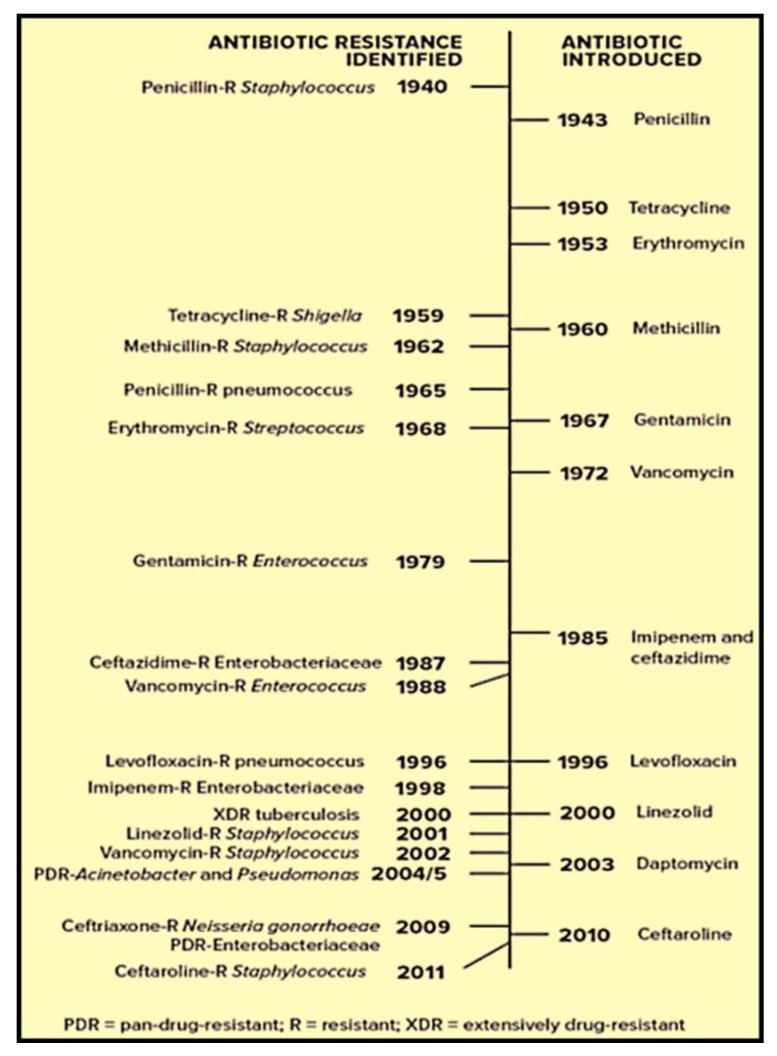
Timeline of events in the development of antibiotic resistance [23].

**Figure 2 microorganisms-08-01934-f002:**
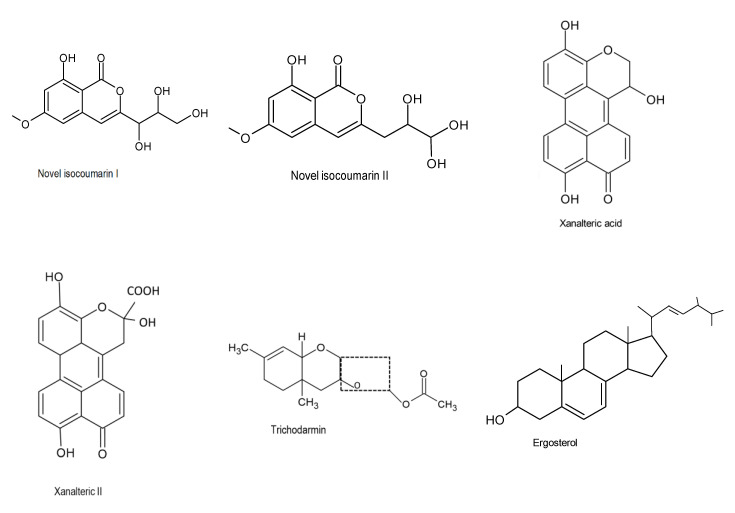
Chemical structures of the bioactive compounds with antibacterial activities [32,33].

**Figure 3 microorganisms-08-01934-f003:**
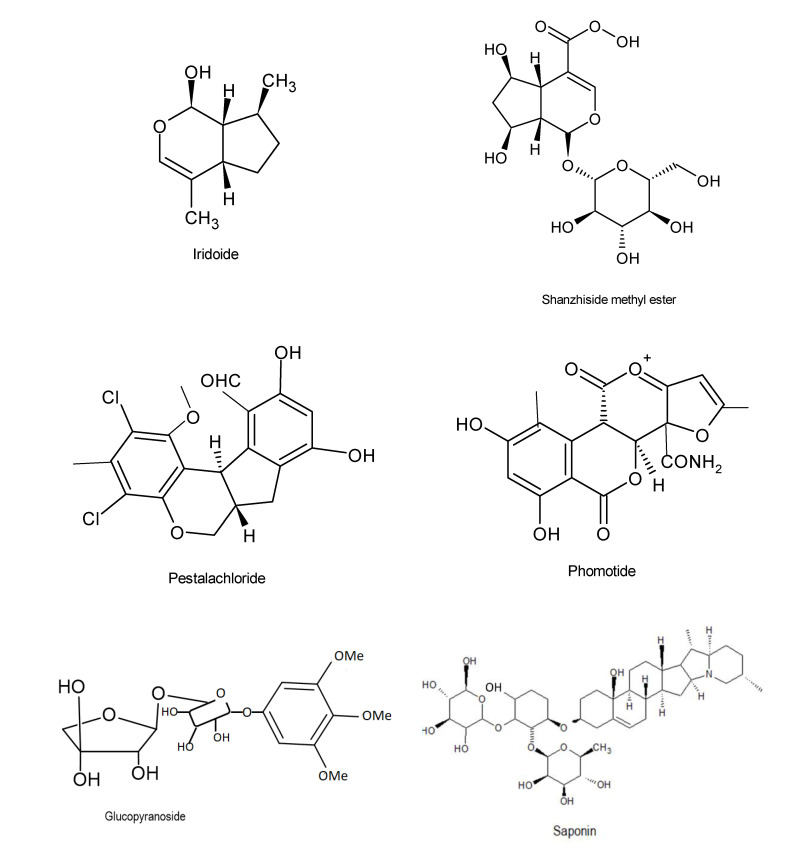
Chemical structures of the bioactive compounds with antifungal activities [33].

**Figure 4 microorganisms-08-01934-f004:**
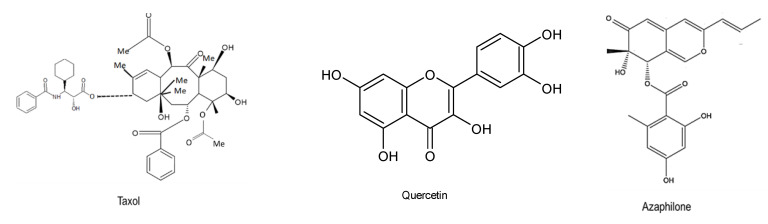
Chemical structures of the bioactive compounds with anticancer activities [25,44,54].

**Figure 5 microorganisms-08-01934-f005:**
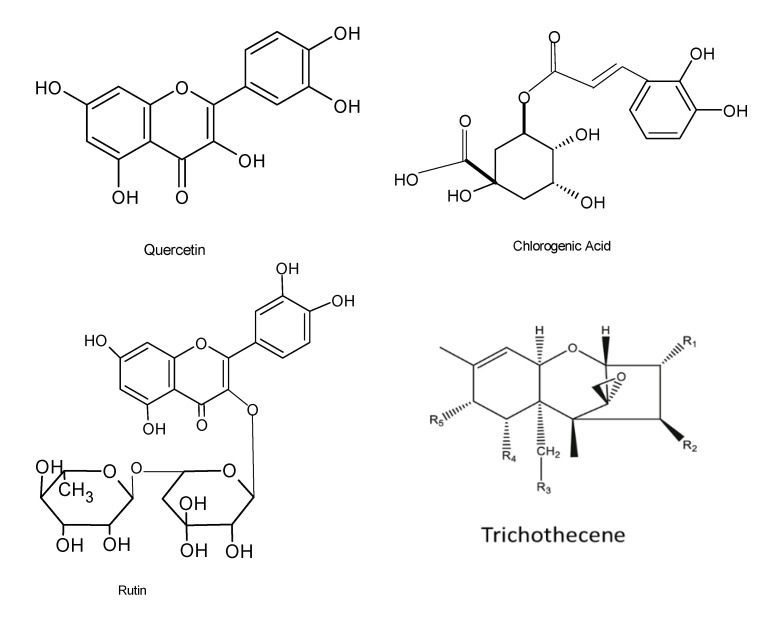
Chemical structures of the bioactive compounds with antioxidant activities [12].

**Figure 6 microorganisms-08-01934-f006:**
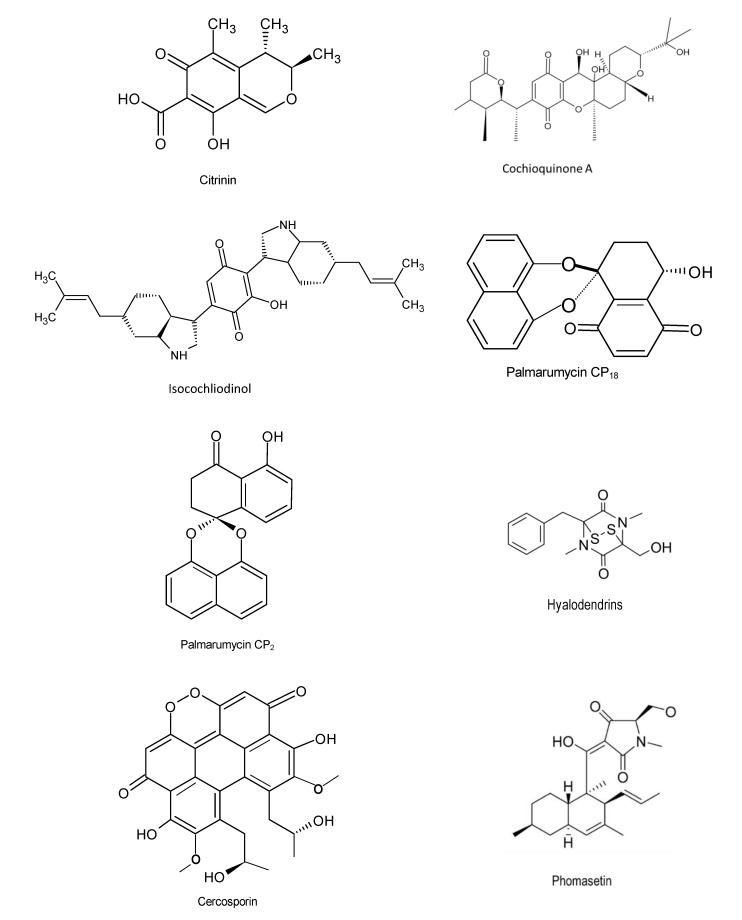
Chemical structures of the bioactive compounds with antiparasitic activities [20].

**Figure 7 microorganisms-08-01934-f007:**
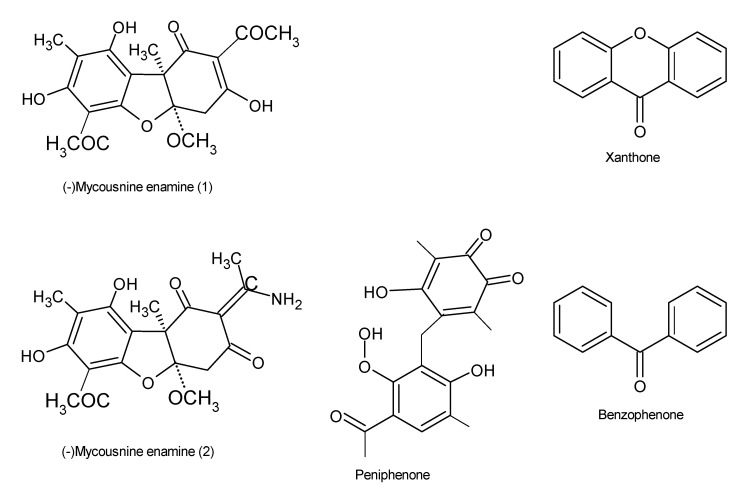
Chemical structures of the bioactive compounds with immunosuppressive activities [99].

**Figure 8 microorganisms-08-01934-f008:**
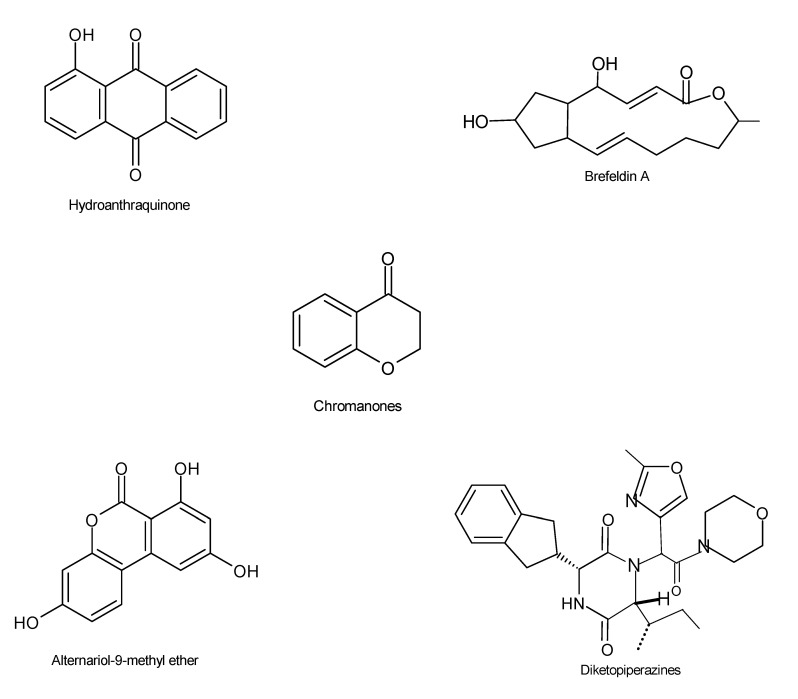
Chemical structures of the bioactive compounds with antiviral activities [12,20,25].

**Figure 9 microorganisms-08-01934-f009:**
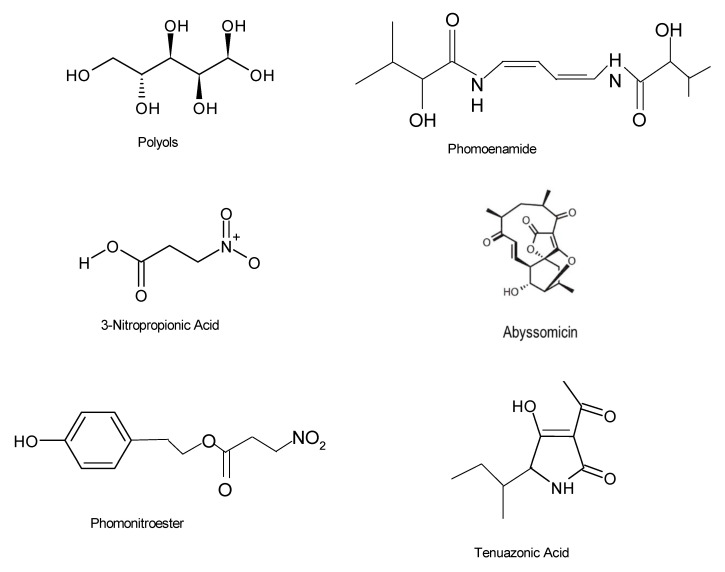
Chemical structures of bioactive compounds with antituberculosis activities [12,20,69].

**Table 1 microorganisms-08-01934-t001:** Summarization of diseases and side effects of oxidative stress.

OXIDATIVE STRESS	
**Neurological**	**Multi-System Effects**
Attention-deficit/hyperactivity disorder (ADHD) Alzheimer’s disease Anxiety and depression Asperger syndrome Autism Multiple sclerosis Parkinson’s disease	Diabetes Cancer Inflammation Fibromyalgia Lyme disease Chronic fatigue syndrome Metabolic syndrome Anxiety Hyperthyroidism Sleep apnea
**Cardiovascular**	**Joints/Skin**
Cardiovascular Disease Angina Pectoris Hypertension Atherosclerosis	Gout Dermatitis Rheumatoid arthritis Carpal tunnel syndrome
**Gastrointestinal Disorder**	**Respiratory**
Crohn’s Disease Gastroesophageal reflux disease (GERD) Gastric ulcers Celiac disease Functional dyspepsia	Chronic obstructive pulmonary disease (COPD) Asthma

**Table 2 microorganisms-08-01934-t002:** List of some bioactive compounds produced by endophytic fungi possessing biological activities.

Endophytic Fungi	Host Plant	Bioactive Compounds	Biological Properties	Activity Level	Ref.
*Penicillium funiculosum* Fes1711 and *Trichoderma harzianum* Fes1712	*Ficus elastica*	Isocoumarin derivatives	Antibacterial activity	MIC = 32 μg/mL	[34]
*Phomopsis* sp. CFS42	*Cephalotaxus fortunei*	Polyketides	Antifungal activity	MIC = 2.5 μg/mL	[44]
*Chaetomium globosum*	*Ginkgo biloba*	Azaphilone alkaloids	Anticancer activity	IC_50_ = 53.4 μM	[58]
*Alternaria alternata* AE1	*Azadirachta indica*	Phenolics and flavonoids	Antioxidant properties	IC_50_ = 38 μg/mL	[28]
*Mycosphaerella nawae* ZJLQ129	*Smilax china*	Amide derivative	Immunosuppressant activity	30 and 300 nM	[98]
*Phomopsis* sp. CGMCC No. 5416	*Achyranthes bidentata*	Chromanones	Antiviral activity	IC_50_ =32.5 μg/ ml	[111]
*Gliocladium sp.* MR41	Culture collection	Polyols	Antitubercular properties	MIC = 3.13 µg/mL	[116]
*Penicillium roqueforti* and *Trichoderma reesei*	*Solanum surattense*	Ferulic acid, cinnamic acid, quercetin, and rutin	Antibacterial activity	MBC = 2.5 µg/mL	[118]
*Lasiodiplodia pseudotheobromae* PAK-7 and *L. theobromae* TN-R-3	*Theobroma cacao* L.	dl-Mevalonic acid lactone, Methyl 6-O-[1-methylpropyl]-á-d-galactopyranoside	Antibacterial activity	MIC= 21 mm	[119]
*Trichoderma asperellum* T1	Culture collection	6-pentyl-2H-pyran-2-one (6-PP)	Antifungal and plant promoting properties	61.31% Inhibition	[120]
*Cladosporium cladosporioides*	*Zygophyllum mandavillei*	3-phenylpropionic acid, 5′-hydroxyasperentin	Antifungal activity	MIC = 15.62 μg/mL	[121]
*Talaromyces purpureogenus*	*Grateloupia filicina*	Talaromyolide K	Antiviral activity	60.11% Inhibition	[122]
*Aspergillus* sp. SCSIO XWS02F40	*Callyspongia* sp.	Asteltoxins	Antiviral activity	IC_50_ = 3.5 μg/mL	[123]
*Diaporthe schini*	*Solanum americanum*	1,4-diaza-2,5-dioxo-3-isobutyl bicyclo[4.3.0]nonane and benzeneethanol	Antioxidant activity	DPPH radical = 96.62%	[124]
*Botryosphaeria dothidea*	Pampa and Atlantic Forest Plants	Hexahydropyrrolizin-3-one and (2-methylpropyl) ester	Antioxidant activity	IC_50_ = 0.206 mg/mL	[125]
*Fusarium solani* S-019	*Camptotheca acuminate*	Camptothecin	Anticancer activity	50 µg/mL	[126]
*Alternaria alternata* KT380662	*Passiflora incarnata* L.	Flavone chrysin (5,7-dihydroxy flavone)	Anticancer activity	IC_50_ = 37.97 μg/mL	[127]
*Diaporthe phaseolorum* 92C	*Combretum lanceolatum*	18-Des-hydroxy Cytochalasin	Antiparasitic activity	IC_50_ = 50 μg/mL	[124]
*Phyllosticta capitalensi*	*Tibouchina granulosa*	Brefeldin and heptelidic acid	Antiparasitic activity	IC_50_ = 50.13 μg/mL,	[89]
*Fusarium solani*	*Glycyrrhiza glabra*	Fusarubin, 3-O-methylfusarubin, and javanicin	Antitubercular activity	MIC = 8 μg/mL	[128]

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
