# Peer review of "Untapped Potentials of Endophytic Fungi: A Review of Novel Bioactive Compounds with Biological Applications"

_microorganisms, 2020, doi:10.3390/microorganisms8121934_

Round 1

Reviewer 1 Report

The authors have worked to improve the quality of the manuscript. However, they still don’t seem to have addressed some corrections that I requested in the first round of review. If the manuscript comes back to me and they’ve not done these corrections, I’m afraid I will reject it once and for all.

  • All species names should be in italic, e.g. Penicillium chrysogenum (line 70) and Taxomyces andreanae (line 73).
  • Streptomyces are BACTERIA, they are NOT fungi. This review is about endophytic fungi. The sentence at lines 602-606 should be removed: “This is therefore evident given that fact that the endophytic fungus Streptomyces TBRC7642 was able to produce eight unknown biosynthesized bioactive compounds, namely abyssomycins Y – Z […]”.
  • The authors have now drawn the chemical structures for figure 2-9, making them look more consistent than before. However, many of the chemical structures that appeared in the original manuscript do not appear in the revised manuscript. I don’t know if this is a formatting error due to the use of track changes, but I can only see newly drawn structures for much fewer molecules than before.

Author Response

The names of all species e.g Penicillium chrysogenum (line 70) and Taxomyces andreanae (line 73) now appear in italics, and this has been done for all others throughout the document. Sections in which the species names were corrected been highlighted in yellow.

We hereby agree that Streptomyces species are indeed bacteria and does not fit within the scope of this review. We have deleted sections/sentences that include Streptomyces species as directed by the reviewer and all changes have been highlighted.

In the corrected manuscript that was resubmitted, we have removed some structures. In this revised version we have drawn and inserted them back in the document. All the chemical structures were drawn using chemSketch software and have been highlighted in yellow. We do apologize for the inconveniences.

Reviewer 2 Report

Despite what authors have replied, I could neither find a deeper analysis of the influence of endophytc fungi on cell cycle and apoptosis nor the suggested reference has been considered (no changes in line 180 are evident to me).

Additionally, I havent asked to remove lines 297-298 but instead to move them to another paragraph.

Please, make the requested changes.

Author Response

The sentence stating “In cancerous cells, phenylpropanoids prevent overexpression of histone deacetylase (HDAC) thus inhibition the cell cycle and inducing apoptosis” has been maintained and a deeper analysis of the influence bioactive compounds produced by endophytc fungi on cell cycle and apoptosis have been included (See Lines 301 – 306). In addition, the suggested reference “Fusarubin and Anhydrofusarubin Isolated from A Cladosporium Species Inhibit Cell Growth in Human Cancer Cell Lines”, Adorisio S. et al, Toxins (Basel). 2019 Aug 29;11(9):503. doi: 10.3390/toxins11090503 has been considered and now appears as reference [69].

The sentence in line 297-298 has been rephrased and now appears as “Despite the fact that root-knot nematodes such as Meloidogyne incognita are persistent parasites in plants and contribute about 5% of global crop losses [32], a biocontrol investigation revealed that the endophytic fungus Piriformospora indica was capable of inhibiting the root-knot nematode parasite thus enhanced plant growth was moved to line 132 under the heading “Bioactive compounds as an alternative antibacterial agents”. With the guidance of the reviewer we have moved the sentence on the basis that it did not fit in the previous heading.

This manuscript is a resubmission of an earlier submission. The following is a list of the peer review reports and author responses from that submission.

Round 1

Reviewer 1 Report

This article describes an updated review on Untapped Wealth of Novel Bioactive Compounds “within”: A Review on Endophytic Fungi possessing Biological Properties. In general, the whole document is very well written, although minor revisions are needed.

General comments

  1. The authors need to include tables summarizing activity values of compounds
  2. Compounds need to be numbered consecutively to avoid repeating the same number through the whole document
  3. The structures of all compounds need to be re-drawn and harmonize in the whole document
  4. IC50 or EC50 need to be changed to IC50 and EC50
  5. The activity of compounds described must include activity values (MIC/IC50/EC50). This will be helpful in appreciating the bioactive potential of metabolites produced by endophytes.

Author Response

Good day

Please find attachment, response to reviewer.

Regards,

Dr Madira Manganyi

Reviewer 2 Report

The present review article deals with bioactive natural products made by endophytic fungi. The topic is definitely of interest to scientists who work on natural products. The manuscript itself is quite informative, however revision is needed before this can be published.

General points on the manuscript:

  • My main issue with the manuscript is that it feels very dry as it mostly is a compendium of bioactive compounds; the authors are just listing molecules isolated from endophytic fungi and telling what their bioactivities are. There is no critical thinking, no outlook or insight as to what strategies researchers who work in this field may want to adopt to discover new bioactive molecules. Therefore, I suggest that the authors expand the present manuscript by adding a section in which they discuss what strategies, or specific groups of plants or their habitats etc., people may use/focus on to discover new bioactive molecules from endophytes.
  • The quality of the language is quite poor and should be improved before re-submission: the authors are strongly advised to seek the help of a proof-reader.

Specific points on the manuscript:

  • All figures include structures of low quality, with different ring and bond sizes and atoms labelled using different font styles, sometimes with numbering on the atoms with no apparent reason. Sometimes R groups are included but there is no mention in the figure or elsewhere of what these may be. The structures seem to have been copied and pasted from other resources, witnessed by the fact that often these are highly pixelated (e.g. top two structures in figure 6) or have a yellow background (e.g. tyrosol in figure 2). The authors should draw all structures using ChemDraw or similar software to ensure that these are all consistent in style and kept in proportion.
  • Latin names should be in italic (e.g. line 60, Penicillium chrysogenum).
  • The authors say in a few occasions that paclitaxel (taxol) is made by endophytic fungi. In reality, there is little to no evidence that fungi can produce paclitaxel. For instance, Heinig et al. (Fungal Diversity (2013) 60:161–170 DOI 10.1007/s13225-013-0228-7) show clearly that pure cultures of the endophytic fungi previously reported to produce paclitaxel are unable to accumulate such molecule, suggesting that paclitaxel might only accumulate in the fungi as a result of the plant’s production and transport into the cells of the microorganisms. The authors are advised to remove any mention of paclitaxel being made by endophytic fungi in their manuscript.
  • Line 45: the authors should clarify the following statement as I struggle to see the link between over propagation and extinction of plants: “However, the over propagation and usage might lead to endangerment and extinction of the plants.”
  • Line 423: Streptomyces are bacteria, not fungi. This sentence should be removed.

Author Response

Good day

Please find attachment for the response to reviewers.

Regards,

Dr. Madira Manganyi

Reviewer 3 Report

The manuscript entitled “Untapped wealth of novel bioactive compounds “within”: a review on endophytic fungi possessing biological properties” is an interesting review that highlights the importance of endophytic fungi in serching for new drugs in various pathological area such as infectious diseases, cancer and others. The review is well designed, however it needs changes in order to be considered for publication.

  1. We strongly suggest an extensive english check;
  2. In the paragraph 2.3, I would eliminate “abnormal cells” in the title and just write “cancer cells”.
  3. In the same paragraph the contribution of cell cycle inhibition and of apoptosis induction by compounds coming from endophytic fungi was not dissected. Please, refer to the following paper to eliminate this vulnus: “Fusarubin and Anhydrofusarubin Isolated from A Cladosporium Species Inhibit Cell Growth in Human Cancer Cell Lines”, Adorisio S. et al, Toxins (Basel). 2019 Aug 29;11(9):503. doi: 10.3390/toxins11090503.
  4. Lines 297-298 should be moved to the paragraph on antibiotics drugs.
  5. The statement that “Current immunosuppressive drugs are ineffective” (line 345) is not correct. There are many effective immunosuppressive drugs (Glucocorticoids, for example). Thus, the sentence should be either eliminated or changed.
  6. The sentence “Those fungal extracts showed activity of less than 30% cytotoxicity in T-lymphocytes” (lines 404-405) should be moved to the paragraph on immunosuppressive activity.

Author Response

Good day

Please find the response to reviewer.

Regards, 

Dr Madira Manganyi

Round 2

Reviewer 2 Report

The manuscript has been considerably improved after revision. However, I am surprised that the authors claim that the chemical structures have been changed or improved.

The chemical structures in virtually all figures are still of very low quality. Some of them are pixelated, some of them have number for atoms and others don't, there is inconsistent use of methyl group representation styles, the structures of different molecules are not in proportion. Please draw the structures using an appropriate software. Then re-submit.

Author Response

The authors utilized ChemSketch software to construct the chemical structures. The whole manuscript was reviewed to improve the language. 
